# Design rules for catalysis in single-particle plasmonic nanogap reactors with precisely aligned molecular monolayers

Gyeongwon Kang [1,2], Shu Hu [1], Chenyang Guo[1], Rakesh Arul [1], Sarah M. Sibug-Torres [1] & Jeremy J. Baumberg [1] ✉

Plasmonic nanostructures can both drive and interrogate light-driven catalytic reactions. Sensitive detection of reaction pathways is achieved by confining optical fields near the active surface. However, effective control of the reaction kinetics remains a challenge to utilize nanostructure constructs as efficient chemical reactors. Here we present a nanoreactor construct exhibiting high catalytic and optical efficiencies, based on a nanoparticle-on-mirror (NPoM) platform. We observe and track pathways of the Pd-catalysed C-C coupling reaction of molecules within a set of nanogaps presenting different chemical surfaces. Atomic monolayer coatings of Pd on the different Au facets enable tuning of the reaction kinetics of surface-bound molecules. Systematic analysis shows the catalytic efficiency of NPoM-based nanoreactors greatly improves on platforms based on aggregated nanoparticles. More importantly, we show Pd monolayers on the nanoparticle or on the mirror play significantly different roles in the surface reaction kinetics. Our data provides clear evidence for catalytic dependencies on molecular configuration in well-defined nanostructures. Such nanoreactor constructs therefore yield clearer design rules for plasmonic catalysis.

Plasmonic gaps constructed from assemblies of metallic nanostructures are an emerging tool for directly studying catalytic reactions[1–4] since they confine optical fields far below the sub-diffraction limit[5–7]. These trapped optical fields trigger photocatalytic reactions with higher efficiency and sensitivity and allow for tracking reaction pathways at molecular scales. Common plasmonic metals such as Au or Ag have been employed for a wide range of plasmonic catalysis applications[8–11] due to their strong electromagnetic field enhancements near the surface. However, these metals are, in many cases, chemically inert, leaving major questions about how to access high chemical selectivity and efficiency. To overcome this, the antenna-reactor concept was developed to combine both plasmonic and catalytic metals in hetero-nanostructures[12–19]. A significant issue has been the imprecision of these nanostructures, with the active surface sites remaining generally unknown.

Here we adopt a precision approach to construction that offers mechanistic understanding. In recent decades it has become possible to modify metallic surfaces by depositing an atomic layer of catalytically active metals such as Pd, Pt, or Cu onto them[20–27]. These metals show surface-sensitive catalytic activities toward specific chemical reactions because of the interaction between d-band electrons and the adsorbates. The deposition method used here is the atomic monolayer coating of metal onto colloidal nanoparticles (NPs) through chemical reduction, to obtain core-shell bimetallic NPs[20,22,27]. Mild-reducing reagents such as ascorbic acid and hydroxylamine hydrochloride are used to reduce metallic anions on the surface of particles and to achieve conformal growth of an atomically thin shell. The second approach adopted is to electrochemically reduce metallic anions onto an electrode surface through underpotential deposition (UPD)[21,23–26]. Atomic monolayer deposition of Pd onto Au surfaces is readily

[1]Department of Physics, Cavendish Laboratory, Nanophotonics Centre, University of Cambridge, Cambridge CB3 OHE, UK. [2]Department of Chemistry, Kangwon National University, Chuncheon 24341, South Korea. ✉e-mail: jjb12@cam.ac.uk

achieved, due to the separation of monolayer and multilayer reduction peaks, as well as the near-matched lattice constant of Pd to Au[26]. We note many other transition metals can be similarly alchemically glazed onto plasmonic nanocavities using these techniques. In comparison to previous concepts using >1 nm-thick Pd structures, the <1 nm coating retains the plasmonic optical properties while transforming the surface chemistry.

The Suzuki–Miyaura C–C cross-coupling reaction is one of the most important chemical reactions catalysed by Pd[28,29]. Traditional reaction studies of C–C cross-coupling have focused on soluble organometallic Pd catalysts with high chemo- and stereo-selectivity[28-32]. Surface-mediated heterogeneous catalytic reactions towards adsorbates bound by chemical bonds offer increased sensitivity and selectivity. Recent efforts to engineer efficient plasmonic catalysis for Suzuki coupling focus on improving catalytic efficiency with different nanostructure assemblies[13-18,33,34]. However, a detailed understanding of the molecular mechanisms and role of nearby surfaces is still missing, so that reaction pathways at the surface are not yet fully understood or controlled.

Here, we demonstrate tuneable plasmonic catalytic reactors to study the Suzuki–Miyaura reaction mechanism of precisely assembled molecular monolayers within nanogaps between facing metal facets at a single particle and a metallic mirror. The chemical activity of the facets coated by a single monolayer of Pd catalyses C–C coupling of surface-bound molecules assembled on the Au mirror substrate. Suzuki–Miyaura reactions are selectively catalysed by the Pd monolayer surface while retaining the plasmonic properties of metallic Au known to provide thermal energy and hot electrons that boost the reaction[1,3,15,35]. These nanoreactors localize optical fields inside the nanogap that both promote the reaction whilst simultaneously tracking the photochemistry using surface-enhanced Raman scattering (SERS) spectroscopy. The SERS molecular fingerprints detect reaction progress in real-time, across a range of nanoreactor designs. Surprisingly, different surfaces show different characteristic "surface" reaction kinetics that involve mass transfer into the nanogap. Implementing a monolayer Pd surface above the molecular layer improves reaction kinetics, whereas Au or Pd surfaces below the molecular layer seem to control mass diffusion by tuning the

polarizability of the nanogap. Such specific design rules for plasmonic photocatalysis improve prospects for applications of plasmonic nanostructures in chemical synthesis and energy conversion.

## Results

### Nanogap reactor preparation

To build robust and well-controlled nanometre-scale reactors, a nanoparticle-on-mirror (NPoM) construct is used where a flat metallic surface is coated with a molecular self-assembled monolayer (SAM) to form the reactant layer. Subsequently, combining decorated NPs and metallic surfaces gives a variety of chemically active nanostructures (Fig. 1). Colloidal Au NPs of 80 nm diameter act as the top photo-catalytic surface. To enhance their catalytic activity, a monolayer of Pd atoms is deposited onto the Au NP surface through the chemical reduction of Pd[2+] in solution, resulting in Au@Pd core-shell NPs (Fig. 1a, see the "Methods" section)[27]. Homogeneous deposition of Pd onto the Au NP surface is confirmed with TEM images and EDS analysis (Supplementary Fig. 1). Comparisons of cyclic voltammograms on Au NPs and Au@Pd NPs with different Pd coverage show from the oxidation peak position (which varies for the first Pd atomic monolayer to subsequent ones because of their different coordination) that deposition of monolayer Pd on Au NPs is achieved as well as on planar Au (Supplementary Fig. 2). Atomically smooth gold surfaces prepared by template-stripping[36] offer well-defined binding sites for thiol SAMs. This gold surface can be pre-coated with a monolayer of Pd atoms through electrochemical UPD (Fig. 1b, see the "Methods" section)[24,26,37]. For the model reaction here, a SAM of 4-bromothiophenol (4-BTP) is adsorbed on the metallic mirror (see the "Methods" section) and then sandwiched inside the NPoM structure to form the plasmonic nanogap reactor (NR). The combination of two different surfaces for NP and mirror results in four different NPoM types (Fig. 1c): Au NP on Au mirror (Au-on-Au), Au NP on Au@Pd mirror (Au-on-Pd), Au@Pd NP on Au mirror (Pd-on-Au), and Au@Pd NP on Au@Pd mirror (Pd-on-Pd). Due to plasmonic field confinement and direct chemical contact between molecules and metal within the nanogap, catalytic reactions within NRs are rapidly triggered under laser illumination. Here the Suzuki–Miyaura C–C cross-coupling of surface-bound 4-BTP is boosted and tracked by 633 nm laser light. We aim to observe the C–C

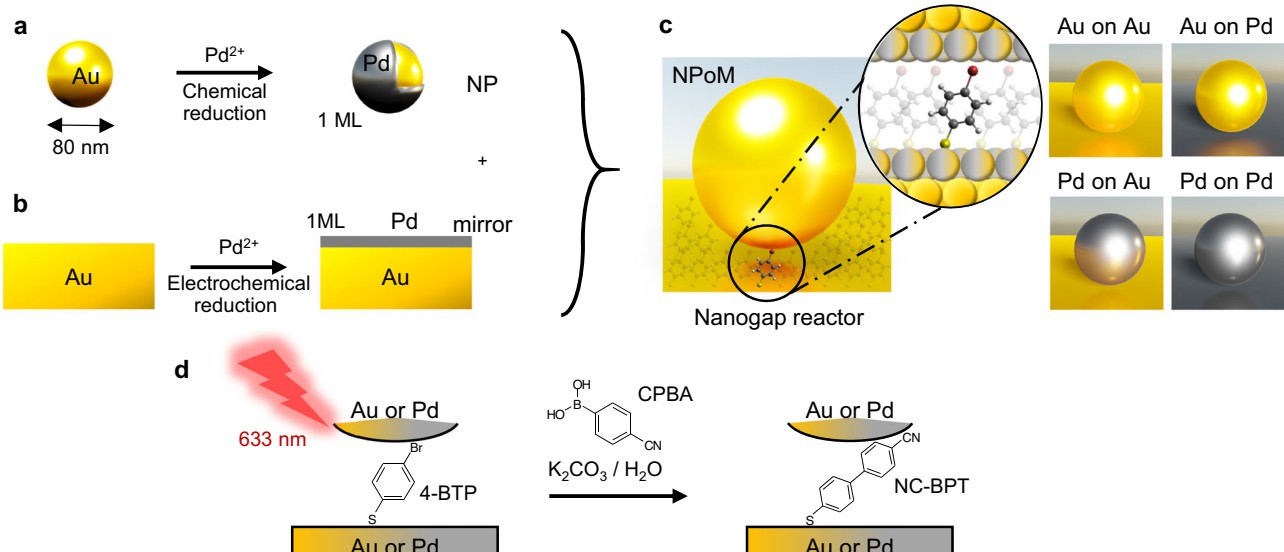

**Fig. 1 | Nanogap reactor construct for Suzuki-coupling catalytic reaction.**
**a** Monolayer (ML) of Pd deposited onto 80 nm Au NPs. **b** ML of Pd electrochemically reduced via UPD onto the mirror of template-stripped Au substrate. **c** Molecular-thick ($d$ = 1.2 nm) nanogap formed by SAM of 4-BTP sandwiched between NP and mirror from (**a**, **b**). Four NR types are Au NP on Au mirror (Au-on-Au), Au NP on Au@Pd mirror (Au-on-Pd), Au@Pd NP on Au mirror (Pd-on-Au), and Au@Pd NP on Au@Pd mirror (Pd-on-Pd). **d** Light-driven Suzuki–Miyaura C–C coupling reaction of 4-BTP and CPBA in the NR, resulting in C–C coupled NC-BPT as a product (Au: yellow, Pd: grey).

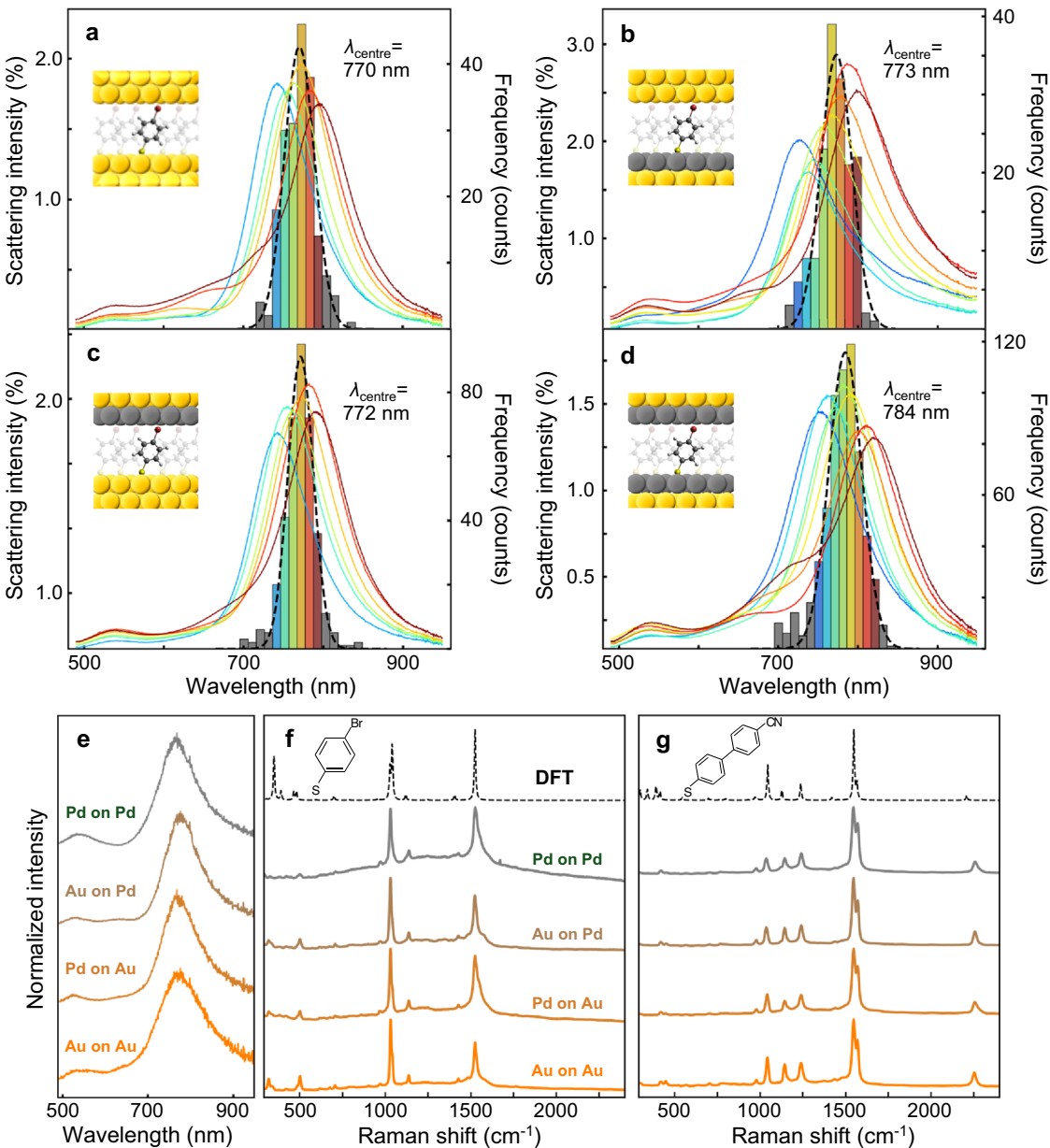

**Fig. 2 | Plasmonic activity of the four NR types.** Histograms of the dominant coupled mode wavelengths from dark-field spectra of >200 NRs and average spectra from each bin of **a** Au-on-Pd, **b** Au-on-Pd, **c** Pd-on-Au, and **d** Pd-on-Pd NRs. Black dotted curves are Gaussian fits of histograms, and the centre wavelength ($\lambda_{centre}$) from each fit is noted at the top. **e** Representative dark-field spectrum of each single NR. Normalized SERS spectra of **f** 4-BTP and **g** NC-BPT measured over 5 min from each NR under ambient conditions without second reactant, for 100 μW laser. From bottom to top: Au-on-Au (orange), Pd-on-Au (darker orange), Au-on-Pd (brown), Pd-on-Pd (grey), and DFT-simulated spectrum of isolated molecule (dashed).

coupling reaction of 4-BTP within the gap with in-diffusing 4-cyano-phenylboronic acid (CPBA) in the presence of base (here $K_2CO_3$) to form the desired surface-bound product (4-cyanobiphenylthiol; NC-BPT, where NC refers to the cyano group) (Fig. 1d)[28–32]. The well-defined molecular monolayer within NPoM constructs enables sim-plified mechanistic analysis at the single-particle scale for an area of <100 × 100 nm².

## Characterization of nanogap reactors

Formation of NRs with different types of NP-mirror is confirmed by dark-field and SEM images (Supplementary Figs. 3 and 4) and white-light scattering spectra, which can identify the wavelength position of the dominant coupled plasmon (Fig. 2a–d). Introducing a Pd mono-layer on the surface of either the NP or the mirror slightly redshifts the coupled mode (by 2–14 nm), most likely due to increases in NP facet

size[38,39]. No significant differences in the dark-field images are observed with and without a Pd ML on the Au surface (Supplementary Fig. 3). Dark-field spectra of representative single NPoMs for each type of NR are spectrally similar (Fig. 2e). The role of the Pd monolayer is insignificant in the optical resonances of the NRs due to its single-atom thickness. This is also confirmed with simulated optical scattering spectra of the NRs using finite-difference time-domain (FDTD) simu-lations (Supplementary Fig. 5). Indeed this is a crucial difference to using either Pd NPs or bulk: the resonant surface plasmon in the NRs penetrates ≈5 nm into the metal, nearly 20-fold further than the Pd ML, thus giving the optics of Au but the surface chemistry of Pd[40]. In order to confirm the plasmonic activity and presence of reactant (4-BTP), SERS spectra are first measured in ambient conditions from different NPoMs without the boronic acid (so no reactions can take place). Averaged SERS spectra of 4-BTP from >20 NPoMs for each type of NR

(Fig. 2f) show SERS bands at 502, 1037, and 1532 cm$^{-1}$, which are characteristic modes of 4-BTP attributed to symmetric stretching between phenyl and Br, symmetric phenyl ring breathing, and phenyl ring stretching vibrations. Likewise, averaged SERS spectra of NC-BPT for each type of NR (Fig. 2g) show characteristic doublet modes at 1550 and 1572 cm$^{-1}$ of the biphenyl and CN modes at 2255 cm$^{-1}$. Experimentally measured SERS spectra match well with density functional theory (DFT)-simulated Raman spectra for both 4-BTP and NC-BPT (details in the "Methods" section) and this confirms the reliable detection of reactant and product molecules in NRs. Stronger peak intensities below 1100 cm$^{-1}$ of 4-BTP are observed with Au NPs, whereas slight peak broadening of SERS bands at 1532 cm$^{-1}$ of 4-BTP and 2255 cm$^{-1}$ of NC-BPT is observed with Pd NPs due to the non-bonding interaction between the polarizable functional group (Br or CN) and Pd surface through a back-donation of $d$ electrons[41–44]. No other significant differences in SERS are observed from the different NRs. Additionally, similar SERS spectral shapes are observed from different NPoMs for the same type of NR, indicating the variation of the dark-field spectra is not directly related to relative SERS peak intensities. Representative SERS spectra from each NPoM further confirm there is no indication of side reactions and photodegradation during 5 min of laser illumination under ambient conditions for both reactant and product (Supplementary Figs. 6, 7). We also note that additional shifted vibrational peaks (known as 'picocavities') or surface-dependent vibrational spectra are not observed, indicating no adatoms or defects are affecting the measurements.

## Tracking reactions with nanogap reactors

The C–C cross-coupling reaction is then initiated by illuminating NRs in a liquid flow cell under an aqueous solution of CPBA along with $K_2CO_3$ (which acts as a hole scavenger). Cyano-functionalized phenylboronic acid is used, unlike previous studies[12,13,16–18,33], to distinguish molecule–NP interactions from molecule–mirror interactions, through the surface sensitivity of the easily-observed CN vibrational mode, which is next to the NP facet. To dynamically track the progress of the reaction, time-series SERS spectra from each NR are recorded (Fig. 3a). The strong SERS amplification of the NPoM construct provides >10$^4$ counts mW$^{-1}$ s$^{-1}$ from the few hundred reactant/product molecules in each nanogap hotspot, enabling short (1 s) exposure times. For the Au-on-Au NR, no product NC-BPT peaks are seen. It is notable that these 4-BTP spectra remain completely stable during 5 min of laser illumination, evidencing that no side reactions can occur and that hot electrons from Au or thermal effects are not directly involved.

By contrast, from all the other NRs, new SERS modes are observed due to the reaction of 4-BTP with CPBA in solution which gives NC-BPT. A comparison between SERS spectra at $t = 1$ s and $t = 5$ min shows the resulting relative SERS peak intensities are strikingly different before/after the reaction for the different NRs (Fig. 3b). Similar spectral shapes of SERS spectra are observed from individual NPoM for the same type of NR, indicating that local surface site effects are negligible in our measurements. We stress that NC-BPT peaks are only observed in the presence of Pd, as expected since the Suzuki–Miyaura reaction is preferred for Pd catalysis. These results convincingly prove the need for Pd in catalysing this reaction and that it can be directly tracked in such precision nanoreactors. Key spectral evidence for the role of the NP surface is the difference in 502 cm$^{-1}$ mode intensity, which DFT shows is a symmetric stretch between phenyl and Br. This peak is much weaker using Au@Pd NPs than with Au NPs, because of the debromination by the Pd surface in the transmetallation step. This further emphasizes the importance of precisely aligned molecular monolayers, since the debromination is much less obvious when using Pd on the mirror surface. For Au-on-Pd NRs, the shift in 1532 cm$^{-1}$ peak is less evident and instead, a shoulder on the 1532 cm$^{-1}$ peak slowly grows along with the 1550 and 1572 cm$^{-1}$ peaks. For Pd-on-Pd the peak is

notably broader than from the other NRs throughout the entire reaction. In this case, the relative intensities of the product peaks (compared to the reactant) are already significantly higher at the beginning of the reaction ($t = 1$ s). This implies the reaction in Pd-on-Pd NRs proceeds immediately upon laser illumination. In contrast, the temporal evolution of SERS during laser illumination of Pd-on-Au NRs shows a clear increase in the C–N peak at 2255 cm$^{-1}$ along with other phenyl vibrations from 1100 to 1300 cm$^{-1}$. Furthermore, the sharp 4-BTP peak at 1532 cm$^{-1}$ rapidly drops in intensity during the ongoing reaction, while two overlapping peaks at 1550 and 1572 cm$^{-1}$ from NC-BPT increase.

Despite this, other adducts are sometimes observed in SERS as a collection of broad C–C modes at 1100–1600 cm$^{-1}$ when using laser powers above 0.2 mW, due to polymerization induced by the excess thermal energy (Supplementary Fig. 8). At such high powers, these adducts are more visible for Au-on-Au NRs than Pd-on-Pd NRs. At low powers, our data instead shows monolayer Pd efficiently converts plasmonic energy from Au into chemical energy at the Pd surface. Four well-distinguished SERS peaks from NC-BPT are observed and are assigned using averaged SERS spectra from >20 NPoMs for each type of NR (Fig. 3c). Averaged SERS spectra of NRs with surface Pd show both reactant (4-BTP) and product (NC-BPT) peaks, and the peaks at 1246, 1550, 1572, and 2255 cm$^{-1}$ are attributed to symmetric phenyl–phenyl stretching, symmetric and asymmetric in-plane deformations, and C–N breathing modes of NC-BPT, respectively (Supplementary Fig. 9). Here we label the 1532 cm$^{-1}$ peak of 4-BTP as $\nu_1$ and 1572 and 2255 cm$^{-1}$ peaks of NC-BPT as $\nu_2$ and $\nu_3$, respectively, for further peak analysis. It is noticeable that relative peak intensities in the 1450–1550 cm$^{-1}$ range are distinctly different for all NRs (blue shaded in Fig. 3c). The intensities of 1550 and 1572 cm$^{-1}$ peaks from NC-BPT are stronger for Pd-on-Au > Pd-on-Pd > Au-on-Pd > Au-on-Au, when compared to the 1532 cm$^{-1}$ 4-BTP peak (Fig. 3d). Different intensities of the C–N mode of NC-BPT at 2255 cm$^{-1}$ reveals a similar trend in relative catalytic activity of NRs. We note this data also confirms that CPBA is able to diffuse into these nanogaps and react, despite the confined geometry. Similar trends in the reaction kinetics are observed for a solution of phenylboronic acid (PBA), indicating the reaction is not affected by the addition of the cyano group (Supplementary Fig. 10).

## Interpretation of SERS spectra

To investigate the detailed kinetics and surface chemistry of different NRs, the SERS peak intensities are analysed over time. The peak analysis is carried out with averaged SERS spectra over multiple NPoMs to disregard any local surface site effects so that the peak intensity trends are only dependent on the type of the surface metals. Each peak of interest is fitted to a Lorentzian curve to yield the peak area. Fitting results are shown in Supplementary Fig. 11 and Supplementary Tables 1–3. The intensity ratio $R_{21}$ of 1572 cm$^{-1}$ ($I_2$) to 1532 cm$^{-1}$ ($I_1$) peaks is used to track the relative reaction progress among NRs. Intensity ratio values are shown in Supplementary Note 1. Surprisingly, the progress of $R_{21}$ varies for each NR type (Fig. 4a, c). First, the initial yield of NC-BPT is quantified from the peaks observed at $t = 1$ s (1st SERS frame). This is most significant for Pd NR surfaces, and this rate doubles for two Pd surfaces in the Pd-on-Pd NR geometry. This is likely because the initial hot carriers generated by the Pd ML surface are fully converted into chemical energy when reactants are precisely assembled into the ideal alignment for reaction. This is not seen in previous reactor-antenna geometries[45–49], because they do not align molecules and metal surfaces with the precision developed here. Moreover, the precise molecular alignment in the nanogap provides direct van-der-Waals contact between Br and Pd atoms that facilitates light-driven oxidative addition (Fig. 4h). Furthermore, this homogeneous molecular alignment yields an oriented gap environment that seems to facilitate diffusion of reactants in solution. Comparison of $R_{21}$ to the intensity ratio of 1550–1532 cm$^{-1}$ peaks show similar trends in the

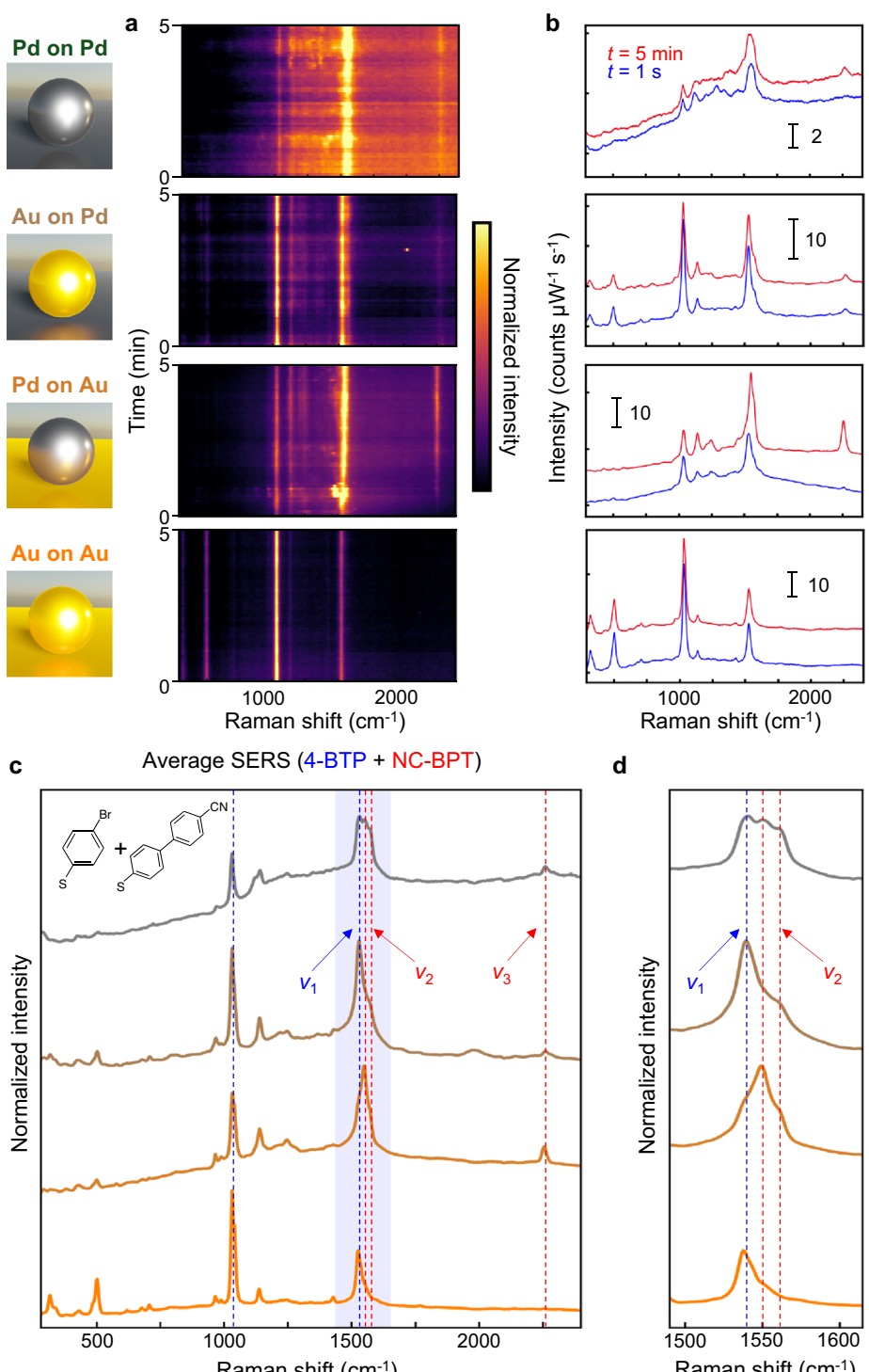

**Fig. 3 | SERS spectra from each NR type loaded with 4BPT in solution of CPBA and K₂CO₃. a** Colourmaps of repeated SERS spectra (300 frames @ 1 s integration times). **b** Initial (*t* = 1 s, blue) and final (*t* = 5 min, red) SERS spectra from colourmaps in (**a**). **c** SERS spectra normalized by the maximum intensity in solution over 5 min for each NR type. **d** SERS spectra of blue-shaded region in (**c**). From bottom to top: Au-on-Au (orange), Pd-on-Au (darker orange), Au-on-Pd (brown), and Pd-on-Pd (grey).

relative peak intensity for all NRs, except that the trend of Pd-on-Au NR is higher than that of Pd-on-Pd NR due to significant spectral overlap between two peaks for Pd-on-Pd NR (Supplementary Fig. 12).

Two distinctively different kinetic behaviours are observed for NRs with either Au- or Pd-coated mirrors. In the former, the intensity ratio initially grows non-linearly before entering a slower linear regime after ≈2 min of laser illumination. By contrast, NRs with a Pd-monolayer-coated mirror show a slow linear growth regime right

from the start. Several efforts to unveil the heterogeneous catalytic mechanism of Suzuki coupling have been reported[50–52]. Among them, the utilization of Au–Pd complex nanostructures to investigate sur-face-'bound reactions suggests detailed mechanistic insights into the reaction[12,15,18]. However detailed discussion on different surface kinet-ics depending on the molecular orientation at different surrounding metal facets at the single-gap scale has been missing, and here is clarified by the precision nano-assembly. To provide an additional

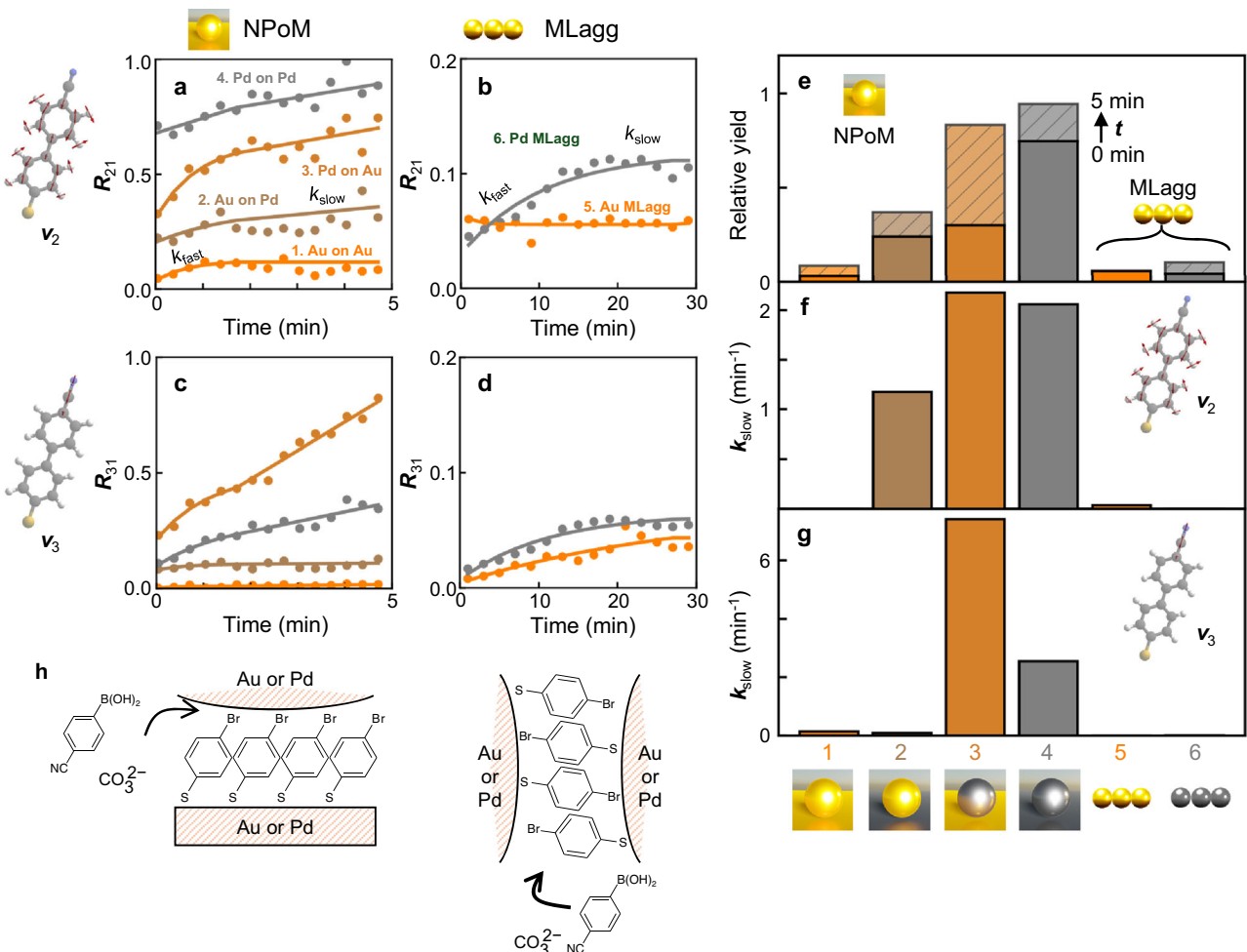

**Fig. 4 | Progress of SERS peak intensity ratio during Suzuki–Miyaura coupling reaction, comparing NRs to monolayer NP aggregates (MLaggs). a, b** Dynamics of peak intensity ratio $R_{21}$ of 1572 cm$^{-1}$ NC-BPT mode ($\nu_2$) to 1532 cm$^{-1}$ mode of 4-BTP ($\nu_1$) for **a** NRs and **b** MLaggs (note multiplied by ×5). **c, d** Peak intensity ratio of 2255 cm$^{-1}$ NC-BPT mode ($\nu_3$) to $\nu_1$ for **c** NRs and **d** MLaggs. **e** Laser-driven catalytic conversion yields at $t = 1$ s (darker bars) and $t = 5$ min (for NRs) or $t = 30$ min (MLaggs) (lighter hatched bars). **f, g** Slow rate constants ($k_{slow}$) fitted from **a–d** using two-step kinetics. **h** Schematic molecular alignment in NPoM and MLagg geometries, showing thiols bound to one or both facet surfaces. Labels 1–6 as defined in (**a, b**).

comparison, a monolayer of aggregated NPs assembled with 4-BTP between the NPs (termed an 'MLagg'[53]) is also prepared, again with and without Pd atomic coatings (details in the "Methods" section and ref. 53). SEM images of Au and Pd MLaggs show the densely packed 2D nanoparticle array with a large number (>20) of nanogaps illuminated within the laser focus area of ≈0.5 × 0.5 mm$^2$ (Supplementary Fig. 13). Note these MLaggs preserve gap sizes of <1 nm[53] which are well coupled with the 633 nm laser (details in Supplementary Note 2)[54]. The SERS spectra of both Au and Pd MLaggs after 30 min of the same laser illumination show significantly (>ten-fold) less catalytic activity compared to NRs (Supplementary Figs. 14, 15), despite similar in-coupling efficiencies. Repeating the analysis on these MLaggs to extract $R_{21}$ (Supplementary Fig. 16 and Fig. 4b, note different axis scales) shows similar dynamics (with fast and slow components) but is surprisingly slower, even with Pd monolayers on both facets. When 785 nm excitation is instead used, almost no reaction is observed (Supplementary Fig. 17), indicating the much greater efficiency of 633 nm excitation in triggering the catalysed reaction. This indicates that the plasmonic coupled mode is not directly related to the catalytic activity in the gap, but instead, lower excitation wavelengths show higher efficiency in triggering the reaction by supplying more optical energy[8,55–57].

This data provides intriguing evidence for unexpected catalytic dependencies on molecular configuration in well-defined

nanostructures, which has been hard to obtain previously[58–60]. Additional information is provided by the appearance of the CN vibration at 2255 cm$^{-1}$ ($I_3$) in both NRs and MLaggs (Fig. 4c, d). This mode is decoupled from other phenyl vibrations below 1600 cm$^{-1}$ and directly measures the diffusion of CPBA into the nanogap where it can react, specifically at the top facet. Unlike the dynamics of $R_{21}$ which increases non-linearly for reactants bound to the Au mirror, the intensity ratio of $I_3$ to $I_1$ ($R_{31}$) increases linearly with time for every NR type, at rates Pd-on-Au > Pd-on-Pd > Au-on-Pd ≈ Au-on-Au (Fig. 4c). Again, in both Au and Pd MLaggs (Fig. 4d), $R_{31}$ increases linearly far slower than in the NRs, with again Pd faster than Au. In all MLaggs, both $R_{21}$ and $R_{31}$ saturate after ≈20 min of laser illumination, suggesting the reaction within these gaps between neighbouring NPs now terminates. In addition, the $t = 0$ product peaks for both Au and Pd MLaggs are significantly smaller than those for NRs, suggesting the reaction energy barrier is much higher for MLaggs than NRs.

## SERS peak fitting

To understand these unusual kinetics and surface dependencies, the dynamics of $R_{21}$ and $R_{31}$ are fit to a two-component kinetic model: (i) an exponential increase at early times and (ii) a linear increase at later times, to extract the relative catalytic yields and rate constants (details in Supplementary Note 3). Note here catalytic yields refer to relative

**Table 1 | Summary of factors affecting C–C coupling reaction kinetics in the nanogaps of NRs and MLaggs**

| Nanogap reactor type | Au-on-Au NR | Au-on-Pd NR | Pd-on-Au NR | Pd-on-Pd NR | Au MLagg | Pd MLagg |
|---|---|---|---|---|---|---|
| Scheme | Fig. 5a | Fig. 5b | Fig. 5c | Fig. 5d | Fig. 5e | Fig. 5f |
| Access to Pd surface | None | 1× | 1× | 2× | None | 2× |
| Pd–Br contact | X | X | O | O | X | O |
| Molecular alignment | O | O | O | O | X | X |
| Gap dipole moment | weak | X | strong | weak | X | X |

(X)O = (in)capable of causing reaction.

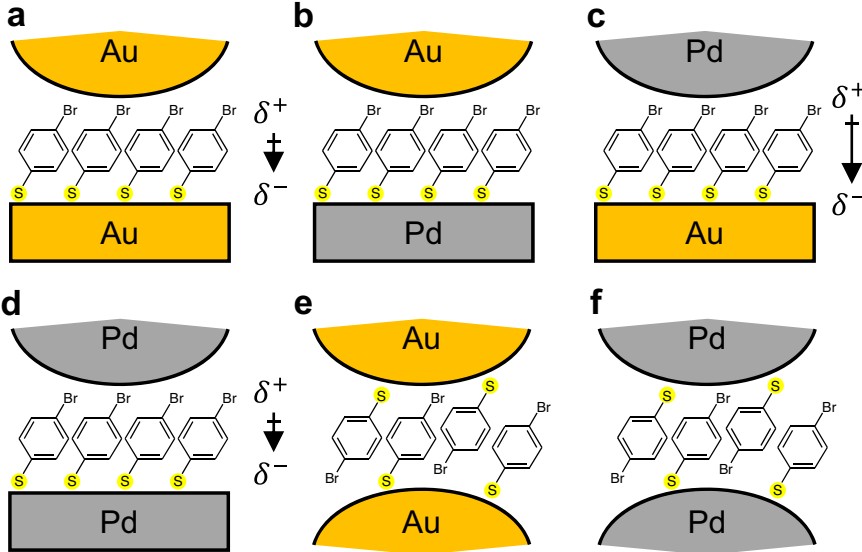

**Fig. 5 | Molecular schematics of NRs and MLaggs.** Schematics show molecular alignments in the nanogap and gap dipole moment strengths of **a** Au-on-Au, **b** Au-on-Pd, **c** Pd-on-Au, **d** Pd-on-Pd NRs, **e** Au and **f** Pd MLaggs.

yields between different reactors, because the reactant/product species are surface-bound. From these fits, the initial and final relative yields, as well as the slow dynamics, are obtained and compared (Fig. 4e–g). As noted above, the precision alignment of molecules between active metal surfaces (in NPoMs) induces an initial reaction faster than the SERS measurement timescale. The surface-mediated catalytic reaction seems to be enhanced by precise molecular alignment (Br-facet alignment) and happens immediately in the presence of excess CPBA. Subsequently, further reaction is possible, controlled by the diffusion of additional CPBA and $CO_3^{2-}$ into the nanogap, and this seems to be faster for Pd NP facets. Here, the role of the diffusion process is emphasized for the surface-bound reaction, which would be totally different from the kinetics of molecules unbound from the surface. Power-dependent measurements reveal that both rates and relative yields increase with increasing power confirming the reaction is optically driven (Supplementary Fig. 18). We suggest that the thiol binding of the 4-BTP to the mirror produces a stable dense SAM that allows diffusion only at the upper NP facet surface (Fig. 4h) so that the CPBA is optimally positioned to react. This is seen for the Pd-on-Au, while for Pd-on-Pd much of the 4-BTP has already reacted after $t = 1$ s. To further understand the role of reactant diffusion into the gap, the number of reactant and product molecules is estimated at the beginning and end of the reaction (Supplementary Fig. 19, Supplementary Note 4, and Supplementary Table 4). From this, we conclude that ≈2–3 molecules s⁻¹ diffuse into the gap of Pd-on-Au NR, which would thus result in ≈24 min saturation time until all reactant molecules inside the gap are consumed. The low initial and final yields in MLaggs, and their slow rates, likely arise from the interleaved attachment of 4-BTP to either facet in each nanogap (Fig. 4h). Allowing the thiols to bind either

way around in the nanogaps clearly gives very different behaviours. Notably, NPoM-based NRs show greatly superior catalytic yields and rate constants than MLaggs.

## Molecular alignment in nanogaps

The surprising catalytic efficiency and the variations in efficiency of these NRs can be considered through a combination of factors (Table 1 and Fig. 5). A well-defined molecular monolayer spacer of 4-BTP allows effective physical contact of the Br to the Pd NP surface, and the 1st-order kinetic process becomes highly efficient. This process seems to be completed even before the end of the first SERS spectra for Pd-coated mirrors (Au-on-Pd and Pd-on-Pd), indicating a relatively fast coupling reaction catalysed by the Pd mirror surface. Subsequent slow rates may originate from the control of molecular diffusion barriers by local polarization. A weak dipole is present within precisely aligned self-assembled molecular layers because the thiol becomes negatively charged. For Au-on-Pd NRs, the molecular dipole is opposed by the dipole moment from an electronegativity (EN) difference between Pd (EN = 2.20) and Au (EN = 2.54). This is further confirmed by calculating the projected density of states (PDOS) using DFT for the Au(111) surface and Pd(111) ML surface on the Au slab (Supplementary Fig. 20 and details in the "Methods" section). Calculation results indicate that the Pd ML surface on Au has an electron-donating character with higher PDOS near the Fermi level. Conversely, the total gap dipole moment is maximized for Pd-on-Au NR when EN and molecular dipoles are added. For Au-on-Au and Pd-on-Pd NRs, this dipole cancels since they have identical but inverted surfaces. The net nanogap dipole seems to affect the mass diffusion of surrounding reactants (CPBA, $CO_3^{2-}$, and water), leading to variable 0th-order kinetics, which is slower than the light-

mediated rates. By contrast, in the nanogaps of MLaggs, all C−C coupling kinetics seem suppressed because no dipole develops due to the randomly oriented 4-BTP layers on both sides of the gap. It is only by comparing catalytic rates in precision nanoreactors that these dependencies can start to be teased apart, offering ways to develop a mechanistic description of photocatalytic reactions and design rules for developing efficient and selective plasmonic photocatalysis.

## Discussion

We note that the mechanism of photocatalytic enhancement for this reaction is not yet well understood, however the literature shows a clear role for plasmonic effects in enhancing the reaction rate beyond thermal effects[13,57,61]. Specifically, the usage of hot-carrier sacrificial agents allows modulation of plasmonic hot-carriers of Au@NP surface which leads to decreased activation energy of the coupling reaction that allows facile reaction at room temperature[61]. Our aim here in precisely defining both the gap morphology and molecular orientation is to show that other effects besides plasmonic heating and hot electrons can significantly alter reaction rates. Further work in this area has thus to proceed by carefully constructing these configurations, which should open up improved ways to prove specific mechanisms.

To conclude, efficient nanogap reactors are developed based on precision plasmonic nanocavity constructs and used to probe the photo-catalytic Suzuki-Miyaura coupling reaction. Plasmonically active Au surfaces are coated with a monolayer of Pd to boost the catalytic activity of these surface nanogap reactors while maintaining their enhanced optical properties. Unusual directional control of the heterogeneous catalytic reaction is demonstrated by assembling four different nanogap reactors with combinations of Au or Pd atomic surfaces on the NP and mirror surfaces. The reaction kinetics of each nanogap reactor reveal a two-step reaction pathway within the confined nanogaps from fast catalytic surface-driven and diffusion-limited processes. We find that surface modification of Au NPs with an atomic monolayer of Pd can give direct contact between the catalytic sites and Br, leading to enhanced reaction rates. Monolayer coatings of Pd onto the Au substrate instead play an important role in polarizing the nanogap to activate the diffusion of the second reactant, base, or hole scavenger. Such control of reaction kinetics and mass diffusion by different nanogap reactor designs are distinguished from traditional plasmonic catalysis experiments due to the facility to control reaction kinetics by selectively aligning specific molecular sites. The significance of nanogap reactors implies a broad range of future applications, including atom-efficient and single-molecule photocatalysis, understanding reactant diffusion in nano-constrained environments, unusual molecular light-emitters, heterogeneous light-driven electrocatalysis and various sensing technologies. These applications can be further expanded towards the design of bulk scale catalysis through metasurface engineering capable of then delivering a broader industrial impact.

## Methods

### Materials

All chemicals are purchased from Aldrich-Merck, unless stated otherwise.

### Preparation of Au@Pd NPs

Au@Pd colloidal NPs are synthesized following ref. 27. with slight modification. 10 mL Au NP colloidal solution with AuNP diameter of 80 nm (BBI Solutions) is first cleaned by centrifuging at $419 \times g$ for several min followed by redispersing the precipitated nanoparticle pellet in 1% sodium citrate (≥99%) aqueous solution. This process is repeated three times. The solution of cleaned Au NPs capped with citrate is cooled in an ice bath for 10 min. 14 μL of 1% ascorbic acid (pharmaceutical primary standard) solution is slowly added to the Au NP solution with thorough stirring, and the solution is then cooled in

an ice bath for 20 min. 10 μL of 1 mM $H_2PdCl_4$ aqueous solution ($PdCl_2$ (≥99.9%) + HCl (37%)) is added to the Au NP solution while shaking thoroughly, followed by 2 min of cooling in an ice bath. This is repeated 10 times to add a total volume of 100 μL of $H_2PdCl_4$ solution and cooled in an ice bath for 2 h. The colloidal solution slowly turns from a dark pink to purple colour upon the reduction of $Pd^{2+}$ onto Au. Monolayer deposition of Pd is demonstrated on the initial AuNPs and characterized using cyclic voltammetry measurements (Supplementary Fig. 2).

### Preparation of Pd ML on Au mirror

A monolayer of Pd is electrochemically deposited onto a template-stripped-Au substrate from 0.1 M $H_2SO_4$ (95-98%) + 0.1 mM $H_2PdCl_4$ aqueous solution using UPD[26]. Pd and Pt wires are used as pseudo-reference and counter electrodes, respectively. The applied potential is swept from 0.2 to 0.02 V vs. Pd with a scan rate of 1 mV s$^{-1}$ to reduce the $Pd^{2+}$. The clear appearance of a reduction peak at 0.05 V vs. Pd before the rise of the overpotential Pd reduction peak at <0 V vs. Pd indicates the deposition of monolayer Pd (Supplementary Fig. 21). The calculated area under the potential sweep peak corresponds to the amount of charge that is indeed predicted to be transferred during the reduction of a single monolayer of (111) Pd surface across the sample.

### Nanogap reactor assembly

Au substrates with, and without the Pd ML are incubated in 2 mM of 4-BTP ethanolic solution for 16 h to form self-assembled monolayers (SAMs) on the surface. Incubated substrates are rinsed with ethanol to remove excess unbound molecules and dried with $N_2$. Au or Au@Pd NP colloidal solutions are centrifuged at $419 \times g$ for 3 min and redispersed in water twice to remove extra capping ligands. 2 μL of NP solution is drop-cast onto SAMs with an additional drop of 10 mM $NaNO_3$ (≥99%) aqueous solution to facilitate the adsorption of NPs onto the hydrophobic 4-BTP SAM surface. The substrates are then rinsed with water immediately and dried with $N_2$.

### NP monolayer aggregate preparation

Monolayer aggregates of Au or Au@Pd NPs are prepared using a liquid−liquid interfacial self-assembly method[53]. 500 μL of NP solution is added to 500 μL of chloroform in an Eppendorf tube. 300 μL of 2 mM 4-BTP ethanolic solution and 10 μL of 1 M $NaNO_3$ aqueous solution are added to initiate the aggregation of NPs. After shaking, the solution is left to settle for another minute, causing the separation of the immiscible chloroform (≥99.8%) and aqueous phases. The aggregated NPs settle to the water−chloroform interface. The upper water/ethanol layer is slowly removed to concentrate the Au aggregate droplet that is formed on the chloroform liquid interface. The droplet is then deposited onto a glass substrate and left to dry. Dried samples are rinsed with excess water and ethanol and dried with $N_2$.

### SERS measurements

All SERS spectra are acquired on a home-built setup based on an Olympus BX51 microscope coupled to a 633 nm diode laser and a white-light source. SERS excitation and collection are achieved through a 633 nm dichroic beam splitter and a ×100 objective (Olympus NA 0.9), and the spectra are recorded by an Andor CCD camera coupled to a Triax 320 spectrometer. A 3D-printed liquid-flow cell is used for all measurements under liquid. The cell is covered with a #1 (thickness = 0.15 mm) coverslip to provide a ≈300 μm-thick liquid layer on top. This design allows high numerical-aperture (NA) SERS measurements with top illumination through glass and liquid layers. 6.7 mM CPBA and 100 mM $K_2CO_3$ aqueous solutions are mixed with a 1:1 volume ratio, and the mixed solution is injected through the flow cell as reactant and base to initiate the coupling reaction with 4-BTP. Substrates are illuminated by white-light through a dark-field

condenser to track each NPoM construct for subsequent SERS measurements with a tight laser focus down to a 0.5 μm spot.

## DFT calculations

Raman scattering cross sections are simulated with DFT using Gaussian 09 software[62]. Full geometry optimization, frequency, and Raman activity calculations are carried out with PBE functionals[63,64] and the 6-311++G(d,p) basis set and the Los Alamos ECP double-z basis set for Au atoms. In the geometry optimization, a thiolate anion of 4-BTP or NC-BPT is bound to an Au atom. The differential Raman scattering cross sections for the $n$th vibrational mode are calculated according to the following equation:

$$\frac{d\sigma}{d\Omega} = \frac{\pi^2}{\epsilon_0^2}(\omega - \omega_n)^4 \frac{h}{8\pi c\omega_n} S_p \frac{1}{45\left(1 - \exp\left(\frac{hc\omega_n}{k_B T}\right)\right)} \tag{1}$$

where $\omega$ is the incident laser frequency (633 nm) and $\omega_p$ and $S_p$ are the frequency and Raman activity of the $n$th mode, respectively. The temperature is assumed to be 298 K, and each Raman peak is broadened with a Lorentzian function with a full width at half-maximum of 10 cm⁻¹. A scale factor of 0.935 for vibrational frequencies is used to better match experimental frequencies.

## PDOS calculations

Spin unpolarized DFT calculations are carried out using the Vienna ab initio simulation package (VASP)[65]. PBE functionals[63,64] are used to describe exchange-correlation effects. The ionic cores are described using the projected augmented wave (PAW) method, with a plane wave up to an energy cutoff of 230 eV. The gold and palladium fcc unit cells are optimized, and the resulting lattice constants are 0.411 and 0.394 nm, respectively, which agree well with experimental values of 0.408 and 0.390 nm. An Au(111) slab structure is built by repeating three layers of (3 × 3) Au unit cell followed by adding 2 nm of vacuum space along the z direction. The top Au atoms are replaced by Pd atoms to obtain the Pd ML surface on Au. For the slab geometry optimization, only the top layer Au or Pd atoms are relaxed while other layers are frozen. The Brillouin zone of slab structures is sampled on a (4 × 4 × 4) Monkorst–Pack $k$-point grid. The same calculation conditions are applied to obtain the PDOS.

## SEM and TEM measurements

SEM images are taken with a FEI Helios NanoLab 650 SEM. TEM images and EDS data are measured with a JEOL JEM-2100FF field-emission TEM.

## Data availability

The data that support the findings of this study are available from the Cambridge Open Data archive[65] and from the corresponding author upon request.

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

## Acknowledgements

We acknowledge support from the European Research Council (ERC) (861950 POSEIDON and 883703 PICOFORCE). G.K. acknowledges support from a 2024 research grant from Kangwon National University and National Research Foundation of Korea (NRF) grant (MSIT No. RS-2024-00342233). S.H. acknowledges funding from the Fundamental Research Funds for the Central Universities (Xiamen University: No. 20720240137).

## Author contributions

G.K., S.H. and J.J.B. conceived and designed the experiments. C.G. carried out SEM imaging experiments. R.A. performed FDTD simulations. MLagg substrates were designed and assembled by S.M.S.-T. G.K. and J.J.B. wrote the manuscript with input from all authors.

## Competing interests

The authors declare no competing interest.
