## [Transparent Peer Review file · Nature Communications]

Design Rules for Catalysis in Single-particle Plasmonic Nanogap Reactors with Precisely Aligned Molecular Monolayers

Corresponding Author: Professor Jeremy Baumberg

This manuscript has been previously reviewed at another journal. This document only contains reviewer comments, rebuttal and decision letters for versions considered at Nature Communications.

Version 0:

Reviewer comments:

Reviewer #1

(Remarks to the Author)

This is a revised version of a manuscript related to the photocatalytic activity of plasmonic nanogaps. The revised version seems to consider the previous comments and addressed them properly.

The work centers on the study of nanoparticles-on-a-mirror systems. Those systems have been explored in plasmonics applications related to surface-enhanced spectroscopy and catalysis. The novelty here is the use of different plasmonic gaps coated with thin layer (monolayer) of catalytic metals, such as Pd. The authors seem to observe different kinetic behaviour from the various systems that were interpreted using kinetic models. The main conclusion is that catalytic nanogaps can be engineered to optimize reactions of industrial relevance. I believe that the manuscript is well-written and provide some interesting insights. I recommend publication of this work after the following minor aspects have been addressed:

- 1) I am surprised to learn that the presence of an atomic layer of Pd does not change the dark field characteristics of the nanogaps. It is well known in plasmonics that molecular adsorption leads to resonance shifts that are used for chemical sensing. My impression is that the presence of a metallic adlayer does provide a change in the dielectric characteristics of the interface that could be comparable to molecular adsorption (although, molecules are obviously larger than single atoms). I suggest that the authors confirm these findings computationally, if possible.
- 2) The identity of the compound formed after the C – C bond formation should also be confirmed either by DFT or by the direct measurement of the compound produced chemically.
- 3) I am also confused by the role of diffusion suggested here. For a nanogap reaction the diffusion should be radial and reach that limit quickly. Considering the number of molecules in the gap, I would expect that saturation would then occur at shorter times than indicated here. Maybe the authors could estimate the number of molecules expected to arrive at the gap per second and the number of molecules in the gap. If the process is totally controlled by diffusion, it might be possible to estimate when the saturation will happen.

Reviewer #2

(Remarks to the Author)

In this work, Baumberg and coworkers constructed four different combinations of nanoreactor (NR) systems with Au and Pd interfaces. The authors drove the Suzuki-Miyaura cross coupling reaction from 4-bromothiophenol (4-BTP) to 4-cyanobiphenylthiol (NC-BPT) through illumination at 633 nm in these NR and compared the reaction efficiency. They observed that Pd-on-Pd or Pd-on-Au yield most products and attributed the higher reactivity to orientation of 4-BTP and diffusion rate of 4-cyanophenylboronic acid (CPBA) into the self-assembled monolayers (SAM) of 4-BTP.

Considering the significance of the Suzuki-Miyaura reaction in chemistry, it is worthwhile to realize the reaction in plasmonic

nanocatalyst systems. However, that is only if the nanocatalyst systems achieve as high yield as organic reaction systems (as Reviewer #1 pointed out) or provide piercing insight into the reaction mechanism. Disappointingly, this work fails to offer both.

Even though the authors employed Au nanoparticles and light activation, I could not find any role of plasmonic activity. No mechanism associated with hot charge carriers or heating is brought about. Note that the Suzuki-Miyaura reaction occurs on Pd surfaces through the redox reaction of Pd ($\text{Pd}^0 \leftrightarrow \text{Pd}^{2+}$) and without heating. It is not clear how hot carriers are involved in the redox reaction of Pd. If the role of AuNPs or Au substrates is limited to physically stand the reactant, 4-BTP, in a form of SAMs, the impact of this work is significantly diminished.

Furthermore, this work lacks rigor that, I believe, is required to be publishable in Nature Communications. The NR systems and molecular orientations that the authors claim play the most significant role in making differences in reactivity must be more rigorously characterized. I find that the authors assert their properties without providing any sound evidence. I am also troubled by the authors' interpretation of the results. The authors often seem sloppy in describing the results and overinterpret them.

For these reasons, I regret to say that I cannot recommend the publication of this manuscript in Nature Communications.

More detailed questions and comments (in addition to the comments by the previous three reviewers) are as follows:

In the first place, I have a strong doubt on the structure of the NR construct. It may not be like what the authors think as described in Fig. 1a and 1b.

1. The authors prepared Au@Pd NPs through reduction of Pd^{2+} using ascorbic acid in colloidal dispersion. How do the authors control the thickness of the shell to a one-atom level (1 ML)? How do they assure that it is uniform? The authors provided cyclic voltammogram (CV) in Fig. S16, but I am not sure how it is exactly correlated to the thickness of the Pd shell. High-resolution transmission electron microscopy (HR-TEM) images must be the direct (and the best) evidence that shows the formation of 1 ML Pd atom shells on AuNPs.
2. Related to 1, what is the surface of Au@Pd like? Citrate is displaced by the Pd atoms (neutral), which should cause aggregation of Au@Pd NPs. If the NPs were sustained in dispersion, it is likely that the AuNP surfaces are not fully covered with Pd atoms.
3. It is a minor point, but core-shell structure is usually denoted as core@shell as the symbol, @, is read "encapsulated by" or "surrounded by". Thus, AuNPs surrounded by the Pd shell must be written as Au@Pd, not Pd@Au that the authors wrote throughout the manuscript.
4. Similarly, the authors prepared Pd-coated Au substrates using an underpotential deposition method and measured CV to claim its ML formation. Have the authors considered atomic layer deposition (ALD) to form a 1 ML-thick Pd layer on Au for sure?
5. The authors claim that the precise (!) molecular orientation of 4-BTP plays an important role in reaction. However, the authors did not provide any experimental data to support it on Au or Pd surfaces. It is well known that thiol molecules form well-defined SAM structures on Au surface, but they are usually alkanethiols that possess strong van der Waals interactions among alkyl chains. The SAM structure of benzene compounds is still controversial. Moreover, the formation and structure of thiols on Pd surfaces are relatively unknown.
6. In the authors' response to the Reviewer #2's comment (3), the authors stated that DF images looked green because of the long-pass filter that was used to measure SERS simultaneously and the image color turned to red when they removed the filter. It doesn't make sense. Long-pass filters must transmit long-wavelength light, so red color should have been observed initially. Please, clarify.

The reaction conditions are not as ideal as described in the manuscript.

7. The authors used citrate-stabilized AuNPs to construct Au-on-Au or Au-on-Pd NR. To adsorb the AuNPs, the authors added NaNO_3 (probably to increase an ionic strength and thereby to reduce the electric double layers of AuNPs, I guess). Have the authors considered the possibility that the citrate ligands or NaNO_3 ions may influence the reactivity? If the hot charge carriers play a role in this plasmon-catalyzed Suzuki-Miyaura reaction, carboxylates on citrates or NaNO_3 ions may serve as hot electron or hole acceptors and consequently interfere with the reaction.
8. Why did the authors use CPBA, instead of phenylboronic acid (PBA) without a CN group. Of course, the CN vibrational peak in the product can be another indicator for the formation of the product, but it adds a complexity in the reaction systems as the CN group can bind to the Au or Pd surfaces. Since the vibrational signature of the biphenyl compound is clearly distinguishable, the authors must run the reaction with PBA.

The overall reaction mechanism.

9. As I mentioned above, it is not clear how the plasmonic effect plays a role in this reaction. The Suzuki-Miyaura reaction is known to occur through the redox reaction of Pd after coordination of halogens. What do the hot electrons do?
10. The first step and one of the most important requirements for the Suzuki-Miyaura reaction is oxidative addition of 4-BTP to Pd. Thus, as Reviewer #2 pointed out (Comment 5), the vibrational peak for Pd-Br or Pd-C should be observed. The authors responded that they did not observe it because of fast kinetics and low Raman scattering cross section. They can hold the reaction simply by blocking the transmetalation step without CPBA. Then, they should be able to detect the Pd-Br stretching mode. Furthermore, the Raman scattering cross section of the Pd-Br stretch should not be that bad. The close location of the bond to the AuNP surfaces must increase the SERS enhancement as well.
11. I am not fully convinced by the authors' model. It's hard to believe that the (relatively) bulky CPBA diffuses into the well-packed (as the authors claim!) 4-BTP SAMs and reacts at the very narrow interface between Pd and 4-BTP.
12. Instead, would it be possible that 4-BTP undergoes hot electron-driven dehalogenation reaction and forms a biphenyl product? The CN Raman peaks may just come from standing-by CPBA molecules near the AuNP surfaces. (The authors should specify what concentration of CPBA was used in the manuscript.)

13. In the response to Reviewer #2's Comment 3 and Lines 243-245, the authors claim that the reaction efficiency is higher when NPoMs are excited at 633 nm than at 785 nm and thus "the dominant coupled mode is not directly involved in the conversion process of optical energy into chemical energy at the surface". Does this indicate that plasmons don't do anything in this reaction? If hot carriers or thermal heating contribute to the reaction, the reaction efficiency must depend on the resonant excitation of coupled plasmon modes. Contrary to the listed papers in the authors' response, there are more papers that support the direct correlation between the resonance condition and plasmonic reaction efficiency.

Some other comments:

14. The authors claim that the plasmon resonance for MLagg lies at 633 nm (Lines 234-237). However, if the gap size is < 1 nm, as the authors claim, and MLagg consists of that many AuNPs, the coupled plasmon mode should be at far longer wavelengths due to the strong coupling.

15. The authors compared the intensities of 1550 and 1572 cm^{-1} peaks for four different NPoM systems in the spectra in Fig. 3c (Line 189). The problem is that the spectra in Fig. 3c are normalized spectra and the authors did not specify to what the spectra were normalized.

16. Please, note that it is customary to place a space between the numeral and the unit. Write "785 nm", instead of "785nm". A few exceptions are %, \$, °. Please, refer to the ACS Style Guide.

Version 1:

Reviewer comments:

Reviewer #1

(Remarks to the Author)

The reviewers have addressed all my concerns. The revised version seems much improved and ready for publication.

Reviewer #3

(Remarks to the Author)

In this manuscript, the authors have systematically studied the impact of Pd monolayers on surface chemical reaction kinetics by constructing various nanoparticle-on-mirror (NPoM) platform. Utilizing Surface-Enhanced Raman Spectroscopy (SERS) technology, combined with multiple characterization techniques, the authors demonstrate that the interaction between molecular configuration and Pd monolayers is the root cause of catalytic reaction dependency. The authors' revisions to the manuscript and their responses to the reviewers' comments have significantly improved the quality of the manuscript.

I believe the work can be accepted after addressing the following issues:

1. How can it be proven that the SERS data in the supporting information Fig.S4&S5 originate from a single NPoM? As the authors previously pointed out, observing a single NPoM might require the spatial resolution of the Raman spectrometer to be within 100 nm. The authors need more evidence to describe the spectral observation of a single NPoM.

2. As the Reviewer #3 mentioned (Comment 9), there is no clear evidence showing that the energy of the Au plasmons transfers to the Pd monolayers. The authors' response indicates that previous studies have demonstrated the role of the Au plasmonic effect in similar structures. It is recommended that the authors cite the source of this conclusion in the manuscript.

3. At $t=0$, the ratio of the R31 peak in the Au&Pd MLagg product (Fig. 4d) is close to 0, but the R21 peak ratio (Fig. 4b) is not close to 0, although both are indeed smaller than those in the NR system. The authors should replace this with an appropriate statement.

4. Demonstrating the existence of the Pd monolayers is crucial for supporting the authors' final conclusions. In the authors' response to the Reviewer #3's comment (1), I believe that relying solely on previous work as evidence is not sufficiently convincing. The authors can provide HRTEM images of the structure in the manuscript (if AC-STEM characterization is available, the evidence would be even more compelling), and present intuitive EDS spectral data to corroborate the presence of the Pd monolayers.

Version 2:

Reviewer comments:

Reviewer #3

(Remarks to the Author)

I have reviewed the earlier version of this paper "Showing how Catalysis in Single-particle Plasmonic Nanogap Reactors depends on Precisely-aligned Molecular Monolayers" submitted by Prof. Baumberg and coworkers. In the rebuttal letter, the more detailed discussion and explanation are provided to further clarify the conceptual and fundamental advances of their work. The quality of this manuscript has been significantly improved after revision and the current manuscript is suitable for publication after addressing a minor issue:

The authors claimed that the position of each NPoM is located according to its dark-field image and synchronized with the laser focus position to demonstrate that the SERS signal originates from a single NPoM. It is recommended to include a

scale bar in the dark-field images (Figure S3), which would make it easier for readers to understand the above statement.

Response to reviewers:

We are delighted that the reviewers are happy '*the revised version seems to consider the previous comments and addressed them properly*', giving '*interesting insights*' (reviewer #1), although reviewer #2 suggests the '*NR systems and molecular orientations...must be more rigorously characterized*'. We thus provide the extensive details they request, which robustly show how mechanistic effects arrive from configuration, as well as revise the manuscript according to all the reviewer detailed comments.

Reviewer #1:

1) I am surprised to learn that the presence of an atomic layer of Pd does not change the dark field characteristics of the nanogaps. It is well known in plasmonics that molecular adsorption leads to resonance shifts that are used for chemical sensing. My impression is that the presence of a metallic adlayer does provide a change in the dielectric characteristics of the interface that could be comparable to molecular adsorption (although, molecules are obviously larger than single atoms). I suggest that the authors confirm these findings computationally, if possible.

> The reviewer is correct that metallic coating might change the dielectric properties of the interface and thus the scattering spectra. We thus carry out FDTD simulations to confirm the effect of a single Pd ML. As in our experimental data, FDTD simulations (Lumerical) show that such a thin adlayer of Pd does not significantly redshift the plasmonic resonance more than $\Delta\lambda \sim 30$ nm, which is comparable to the inherent spread in measured dark-field NPoM resonances (due to $\pm 5\%$ differences in facet size, shape, and nanoparticle diameter). This is now shown in a new SI Fig. S3a, where only small changes in resonance wavelength and scattering intensity are seen from FDTD when coating with a Pd adlayer. The near-field strength ($\text{Re}\{E_z\}$) is also little affected (SI Fig. S3b). Separate ab-initio calculations confirm this result [along the lines of Nano Lett. (2023); DOI: 10.1021/acs.nanolett.3c02537]. The reason that a monolayer of Pd has such small effect is the larger 5nm ($\sim 17\text{ML}$) penetration depth of the plasmon, which thus still sees mostly Au.

2) The identity of the compound formed after the C–C bond formation should also be confirmed either by DFT or by the direct measurement of the compound produced chemically.

> Reference SERS spectra of this product are presented in Fig. 2g. A comparison between spectra of the product and reference confirms the formation of NC-BPT due to C-C bond formation. We also confirm the SERS spectra by comparing with the calculated Raman spectra using DFT as suggested. The simulated Raman spectra match well with the experimental ones as shown in Fig. 2f and 2g.

3) I am also confused by the role of diffusion suggested here. For a nanogap reaction the diffusion should be radial and reach that limit quickly. Considering the number of molecules in the gap, I would expect that saturation would then occur at shorter times than indicated here. Maybe the authors could estimate the number of molecules expected to arrive at the gap per second and the number of molecules in the gap. If the process is totally controlled by diffusion, it might be possible to estimate when the saturation will happen.

> The reviewer suggestions help us quantitatively analyze diffusion dynamics. We use $f = w/D = 0.46$ to obtain the relative facet size (w) with respect to the nanoparticle diameter (D) for a $D = 80\text{nm}$ NPoM [ACS Photonics 2022, 9, 2643]. This results in an area of $1,060\text{ nm}^2$ under the NP facet ($w=37\text{nm}$). The common surface adsorption superlattice structure of well-packed

thiolates is $c(4 \times 2)$ [Langmuir 2007, 23, 12208; Phys.Chem.Chem.Phys. 2014, 16, 19017; RSC Adv. 2014, 4, 27730]. Based on this adsorption structure, we obtain ~4,900 molecules located within a gap on a fcc(111) surface.

From our DFT calculations, the Raman intensity of the NC-BPT 1572cm^{-1} mode is 23% stronger than that of 4-BTP 1532cm^{-1} mode. Using this, we can convert the intensity ratios shown in SI Sections S1-2 into populations of reactant and product. From this, we then also calculate the radial diffusion length from the edge of the NP facet to the inner unreacted molecules. The details and calculated values are now included in Section S4 and Fig. S17. This estimate suggests that 2.6 molecules per second need to diffuse into the gap, for the Pd-on-Au NR. Therefore the saturation time (time for all reactant molecules in the gap to be consumed) is expected to be ~24min.

Reviewer #3:

Considering the significance of the Suzuki-Miyaura reaction in chemistry, it is worthwhile to realize the reaction in plasmonic nanocatalyst systems. However, that is only if the nanocatalyst systems achieve as high yield as organic reaction systems (as Reviewer #1 pointed out) or provide piercing insight into the reaction mechanism. Disappointingly, this work fails to offer both.

> The key advance presented in this paper is indeed related to mechanism, showing that the reaction rate depends on the configuration of the molecules at the surface. Since this has not been controlled in previous work, which found it hard to control metal and molecular architecture, this is an important contribution, as noted by the other reviewers. Indeed the useful comments from this reviewer give further details of such alignment, as discussed below.

Even though the authors employed Au nanoparticles and light activation, I could not find any role of plasmonic activity. No mechanism associated with hot charge carriers or heating is brought about. Note that the Suzuki-Miyaura reaction occurs on Pd surfaces through the redox reaction of Pd ($\text{Pd}^0 \leftrightarrow \text{Pd}^{2+}$) and without heating. It is not clear how hot carriers are involved in the redox reaction of Pd. If the role of AuNPs or Au substrates is limited to physically stand the reactant, 4-BTP, in a form of SAMs, the impact of this work is significantly diminished.

> The main plasmonic effect is the extreme enhancement of light in the nanogaps, which increases the local intensity by $>10^5$ and thus all related hot-electron effects. Indeed there are investigations on the detailed role of plasmonic effects for Au@Pd surfaces on C-C coupling reactions [eg refs 13,59,63]. The dependence on reaction rates and laser powers in those examples confirm the reaction is further boosted by plasmonic effects on top of thermal effects. In point #9, we provide more detailed comments on the role of hot carriers.

During our SERS measurements, NPoMs on the same substrate are simultaneously exposed to the reactant solution, but illuminated at different times after this. However we find the reaction progress of different NPoMs is unrelated to its time before light illumination, indicating that evident thermal energy barriers exist for the reaction to be initiated within each nanogap. The reviewer is correct that the exact reaction mechanism is under significant debate, but that is not what we focus on here, see discussion below.

In the first place, I have a strong doubt on the structure of the NR construct. It may not be like what the authors think as described in Fig. 1a and 1b.

1. The authors prepared Au@Pd NPs through reduction of Pd²⁺ using ascorbic acid in colloidal dispersion. How do the authors control the thickness of the shell to a one-atom level (1 ML)? How do they assure that it is uniform? The authors provided cyclic voltammogram (CV) in Fig. S16, but I am not sure how it is exactly correlated to the thickness of the Pd shell. High-resolution transmission electron microscopy (HR-TEM) images must be the direct (and the best) evidence that shows the formation of 1 ML Pd atom shells on AuNPs.

> The monolayer coating strategy we adopt in this work is a mature method. Indeed, several papers already show the formation of a uniform Pd monolayer using HR-TEM (Fig.R1) [J.Am.Chem. Soc. 2010, 132, 12480; Cell Rep.Phys.Sci. 2022, 3, 101105; Chem.Eur.J. 2012, 18, 8150].

Figure Redacted

2. Related to 1, what is the surface of Au@Pd like? Citrate is displaced by the Pd atoms (neutral), which should cause aggregation of Au@Pd NPs. If the NPs were sustained in dispersion, it is likely that the AuNP surfaces are not fully covered with Pd atoms.

> It is known that citrate is not displaced by Pd atoms - instead Pd²⁺ is only reduced when it reaches the metal surface, since the ion is small enough to diffuse through the citrate layer. The solution has excess citrate during synthesis, so the surface is quickly occupied by citrate. As long as citrate caps the nanoparticles, aggregation does not depend on the Pd adlayer coverage, and we do not ever observe any evidence of aggregation. This is related to well-known ligand exchange mechanisms during the growth of metal adlayers on ligand-protected Au NPs [J.Am.Chem.Soc. 2018, 140, 118998; Commun.Chem. 2022, 5, 71].

3. It is a minor point, but core-shell structure is usually denoted as core@shell as the symbol, @, is read "encapsulated by" or "surrounded by". Thus, AuNPs surrounded by the Pd shell must be written as Au@Pd, not Pd@Au that the authors wrote throughout the manuscript.

> As the reviewer suggests, we now denote Au core/Pd shell structures as Au@Pd in the manuscript.

4. Similarly, the authors prepared Pd-coated Au substrates using an underpotential deposition method and measured CV to claim its ML formation. Have the authors considered atomic layer deposition (ALD) to form a 1 ML-thick Pd layer on Au for sure?

> Indeed this is something we have utilized previously for spacer layers (DOI: 10.1038/s41928-020-00478-5), but not for metals. ALD is indeed a powerful technique to

achieve highly uniform molecular monolayers. However ALD requires an oxide (or hydroxyl) surface and functionalized precursors to achieve specific chemical functionalization. This would significantly decrease SERS activity in our work as it prevents reactant diffusion and binding. Additional annealing steps would also intermix metals such as Pd and Au.

5. The authors claim that the precise (!) molecular orientation of 4-BTP) plays an important role in reaction. However, the authors did not provide any experimental data to support it on Au or Pd surfaces. It is well known that thiol molecules form well-defined SAM structures on Au surface, but they are usually alkanethiols that possess strong van der Waals interactions among alkyl chains. The SAM structure of benzene compounds is still controversial. Moreover, the formation and structure of thiols on Pd surfaces are relatively unknown.

> Indeed alkanethiols have been well studied on Au, but aromatic thiols are also very well characterized. The reactant and product thiols studied in our work (4-BTP and NC-BPT) both possess aromatic phenyl rings which have even stronger intermolecular interactions due to coupling between π electrons. Numerous studies report well-defined SAM structures of such aromatic thiols on Au surfaces using multiple techniques [Langmuir 2001, 17, 95; Langmuir 2001, 17, 2408; J.Phys.Chem.B 2001, 105, 4058; Langmuir 2002, 18, 2717].

There are many recent efforts that investigate the structure of thiol SAMs on Pd, due to the importance of tuning catalytic selectivity of Pd by blocking active sites using thiols. We suggest a few examples reporting well-defined structures of thiol SAMs on Pd both experimentally and theoretically [J.Am.Chem.Soc. 2003, 125, 2597; Langmuir 2008, 24, 10838; Nat.Mater. 2010, 9, 853; RSC Adv. 2014, 4, 27730].

6. In the authors' response to the Reviewer #2's comment (3), the authors stated that DF images looked green because of the long-pass filter that was used to measure SERS simultaneously and the image color turned to red when they removed the filter. It doesn't make sense. Long-pass filters must transmit long-wavelength light, so red color should have been observed initially. Please, clarify.

> We appreciate the reviewer's eagle-eye. We correct our previous response: a shortpass dichroic mirror is used to maximize SERS efficiency (Thorlabs DMSP638R), hence giving green DF images.

The reaction conditions are not as ideal as described in the manuscript.

7. The authors used citrate-stabilized AuNPs to construct Au-on-Au or Au-on-Pd NR. To adsorb the AuNPs, the authors added NaNO_3 (probably to increase an ionic strength and thereby to reduce the electric double layers of AuNPs, I guess). Have the authors considered the possibility that the citrate ligands or NaNO_3 ions may influence the reactivity? If the hot charge carriers play a role in this plasmon-catalyzed Suzuki-Miyaura reaction, carboxylates on citrates or NaNO_3 ions may serve as hot electron or hole acceptors and consequently interfere with the reaction.

> To avoid possible influence of ions on the reaction, the samples are thoroughly washed with water after NPoM formation. Our group has reported several studies on the role of ligands within nanogaps. Specifically, formation of surface adatoms is extensively studied using SERS during NP aging by citrate [ACS Nano 2020, 14, 6889]. We report that citrate reconfiguration is observed on the scale of a few days, or under harsh chemical conditions such as high temperature or low pH. More importantly, the influence of adsorbed ligands is detected from unusual 'picocavity' spectra or multiple SERS peaks within the vibrational range 1000-1600 cm^{-1} .

1. We note in the paper that such spectral behavior is only observed at high laser powers (Fig. S6). This indicates C-C coupling reaction selectively proceeds without any interference of other organics/radicals at low laser powers. This is likely due to the high energy barriers for other reaction pathways such as electrochemical bias or extreme pH conditions. (e.g. NO_3^- reduction or radical reaction; J.Phys.Chem.C 2023, 127, 5425)

8. Why did the authors use CPBA, instead of phenylboronic acid (PBA) without a CN group. Of course, the CN vibrational peak in the product can be another indicator for the formation of the product, but it adds a complexity in the reaction systems as the CN group can bind to the Au or Pd surfaces. Since the vibrational signature of the biphenyl compound is clearly distinguishable, the authors must run the reaction with PBA.

> As the reviewer suggests, we also carry out SERS measurements with PBA and include the results in the SI Fig. S8. Here we use a higher concentration of 67mM (10x higher than CPBA concentration) to observe saturation of the reaction within a short amount of time. As expected, similar temporal changes in SERS are observed for all types of NR. Due to the higher reactant concentration, product SERS peaks grow faster than the reaction under CPBA solution. This shows the CN indicator group does not modify significantly the reaction.

The overall reaction mechanism.

9. As I mentioned above, it is not clear how the plasmonic effect plays a role in this reaction. The Suzuki-Miyaura reaction is known to occur through the redox reaction of Pd after coordination of halogens. What do the hot electrons do?

> The reviewer is correct that the C-C coupling reaction is dominantly catalyzed by the redox of Pd. Again some literature reports the plasmonic effects of Au@Pd surfaces on C-C coupling reactions [eg refs 13,59,63]. Specifically, the roles of hot electrons/holes on the Au@Pd NP surface is well studied (see Fig.R2). In this example, we learn that both hot electrons and holes should be provided to progress the reaction on a Pd surface. The main goal in our paper is to study the critical role of precise nanogap alignment through novel surface engineering. Unfortunately, revealing the detailed role of hot electrons requires a completely new experimental design, which is planned for future studies opened up by our approach.

Figure Redacted

10. The first step and one of the most important requirements for the Suzuki-Miyaura reaction is oxidative addition of 4-BTP to Pd. Thus, as Reviewer #2 pointed out (Comment 5), the vibrational peak for Pd-Br or Pd-C should be observed. The authors responded that they did

not observe it because of fast kinetics and low Raman scattering cross section. They can hold the reaction simply by blocking the transmetalation step without CPBA. Then, they should be able to detect the Pd-Br stretching mode. Furthermore, the Raman scattering cross section of the Pd-Br stretch should not be that bad. The close location of the bond to the AuNP surfaces must increase the SERS enhancement as well.

> Considering the reviewer's comment carefully, we now identified every Raman mode of 4-BTP with DFT. Indeed, the SERS peak at 502cm^{-1} (seen in Fig. S7) is due to the breathing between phenyl ring and Br. We note that this peak exists regardless of the NR type in Fig. 2f without reactant solution. On the other hand, NC-BTP SERS spectra do not show the characteristic Br breathing mode in Fig. 2g, experimentally confirming the mode originates from 4-BTP. As the reviewer suggests, tracking this mode during the reaction with reactant solution helps us better understand the transmetalation step. In Fig. 3a and b, the Br breathing mode has relatively strong intensities at the Au NP surface, whereas it completely disappears on the Au@Pd NP surface. This is a clear indication of transmetalation upon the dissociation of Br by the Pd surface. In addition, it confirms that the molecules within the gap are aligned so that the Br atom touches the Pd surface above the SAM. We now emphasize this in the manuscript in more detail.

11. I am not fully convinced by the authors' model. It's hard to believe that the (relatively) bulky CPBA diffuses into the well-packed (as the authors claim!) 4-BTP SAMs and reacts at the very narrow interface between Pd and 4-BTP.

> Thiol SAMs are known to be in a gel-like state, and adsorbed molecules can move between nearby surface sites. This dynamic behavior creates transient spaces for CPBA to diffuse into the gap. Indeed we see such processes with inert thiol SAMs, where immersion of the NRs in a second molecular thiol solution replaces at least 50% of the first SAM from the gap (Fig.R3).

Fig.R3: Schematics and SERS of NRs produced from initial SAMs of BPT or TPT, compared to initially BPT followed by 5 min immersion in TPT showing the replacement of >60% BPT (unpublished).

Such processes can be promoted by thermal gradients from plasmonic confinement heating effects and enhanced gap dipole effects, as noted in the manuscript. In addition, this kinetics can be enhanced by any disruption of the SAM structure from formation of a larger product molecule. We note previous papers from our group confirm molecular diffusion into nanogaps during electrocatalytic reactions [Nat.Catal. 2021, 4, 157], and photoreactions [Nano Lett 2013, 13, 5985].

We now quantitatively estimate the ratio of 'reacted' to 'unreacted' area within the nanogap in response to reviewer #1 point 3 (Section S4). Considering that the diffusion-limited

process mainly develops from the annular edge of the circular gap, the required diffusion length is not particularly large. Therefore, our observations are reasonable showing the reactant molecules can slowly diffuse in from the edge of the NP facet.

12. Instead, would it be possible that 4-BTP undergoes hot electron-driven dehalogenation reaction and forms a biphenyl product? The CN Raman peaks may just come from standing-by CPBA molecules near the AuNP surfaces.

> The resulting biphenyl, 4,4-biphenyldithiol (BPDT), has been well-studied before, and has SERS known to have strong peaks at 1160 and 1560 cm^{-1} . We do not observe any SERS fingerprints of BPDT during the reaction, which fully rules out the possibility of dehalogenation followed by dimerization.

(The authors should specify what concentration of CPBA was used in the manuscript.)

> The concentration of CPBA is mentioned (6.7 mM) in the manuscript in the Methods – SERS measurements section, but we now include this in the main text as suggested.

13. In the response to Reviewer #2's Comment 3 and Lines 243-245, the authors claim that the reaction efficiency is higher when NPoMs are excited at 633 nm than at 785 nm and thus "the dominant coupled mode is not directly involved in the conversion process of optical energy into chemical energy at the surface". Does this indicate that plasmons don't do anything in this reaction? If hot carriers or thermal heating contribute to the reaction, the reaction efficiency must depend on the resonant excitation of coupled plasmon modes. Contrary to the listed papers in the authors' response, there are more papers that support the direct correlation between the resonance condition and plasmonic reaction efficiency.

> In this NPoM geometry, the near-field enhancements are very strong ($E/E_0 > 100$) across the entire visible range, with additional enhancements at multiple resonances [ACS Photonics (2022); DOI: 10.1021/acsp Photonics.2c00116]. Many of these are not seen in scattering spectra as they reradiate poorly. Our statement merely noted that the dominant (10) mode is not seen to overpower other effects in this reaction rate.

As the reviewer notes, there is much controversy about how plasmonic driving works in this reaction, and we emphasize that this is not our focus (as they say, many papers have contradictory results). Instead we show how other configurational features can change the reaction rates, and thus all work in this area should much more carefully control all such configurational factors. The reviewer suggestions are helpful and we thus add a discussion about this.

14. The authors claim that the plasmon resonance for M_Lagg lies at 633 nm (Lines 234-237). However, if the gap size is < 1 nm, as the authors claim, and M_Lagg consists of that many AuNPs, the coupled plasmon mode should be at far longer wavelengths due to the strong coupling.

> The reviewer is correct that the longest wavelength coupled plasmon mode appears at $\sim 1.4 \mu\text{m}$ when coupling many 1nm gaps between 80nm Au NPs. However DF spectra of Au M_Laggs show the presence of higher order plasmonic modes around 633nm regardless of NP size (Fig. R3 grey arrows). We thus clarify our description in the main text.

Fig.R3: Scattering spectra of 1ML films with different Au NP diameters (from Fig. 1d of [55]). Arrows mark resonance positions.

15. The authors compared the intensities of 1550 and 1572 cm^{-1} peaks for four different NPoM systems in the spectra in Fig. 3c (Line 189). The problem is that the spectra in Fig. 3c are normalized spectra and the authors did not specify to what the spectra were normalized.

> This is indeed important. Each spectrum in Fig. 3c is normalized by the maximum SERS intensity. We now clarify this in the manuscript.

16. Please, note that it is customary to place a space between the numeral and the unit. Write “785 nm”, instead of “785nm”. A few exceptions are %, \$, °. Please, refer to the ACS Style Guide.

> As the reviewer suggests, we now space the numerals and units throughout.

Response to reviewers:

We are delighted that reviewers #1 and #2 are now completely happy with the revised manuscript, and that reviewer #3 accepts we “*demonstrate that the interaction between molecular configuration and Pd monolayers is the root cause of catalytic reaction dependency*” and that the “*revisions .. and responses to the reviewers' comments have significantly improved the quality of the manuscript*”, which they note “*can be accepted after addressing the following issues*”. We thus respond to their detailed questions.

Reviewer #3:

1) How can it be proven that the SERS data in the supporting information Fig.S4&S5 originate from a single NPoM? As the authors previously pointed out, observing a single NPoM might require the spatial resolution of the Raman spectrometer to be within 100 nm. The authors need more evidence to describe the spectral observation of a single NPoM.

> NPoMs are sparsely dispersed across the surface as shown in dark-field images in Fig. S2. A tight <1 μ m diffraction-limited laser focus on each NPoM is confirmed by CCD camera images. The position of each NPoM is located according to its dark-field image, and then synchronized with the laser focus position. We confirm that no SERS signal is observed without NPs on the film, and that it also disappears if the laser is slightly laterally displaced off an NPoM. This shows we measure only a single NPoM at a time.

2) As the Reviewer #2 mentioned (Comment 9), there is no clear evidence showing that the energy of the Au plasmons transfers to the Pd monolayers. The authors' response indicates that previous studies have demonstrated the role of the Au plasmonic effect in similar structures. It is recommended that the authors cite the source of this conclusion in the manuscript.

> This is a useful suggestion. We now cite prior studies demonstrating the role of Au plasmonic effects. See the last paragraph of page 10, summarizing the results [refs 13, 59, 63].

3) At $t=0$, the ratio of the R31 peak in the Au&Pd MLagg product (Fig. 4d) is close to 0, but the R21 peak ratio (Fig. 4b) is not close to 0, although both are indeed smaller than those in the NR system. The authors should replace this with an appropriate statement.

> We correct this statement as suggested, seen in the highlighted manuscript on page 9.

4) Demonstrating the existence of the Pd monolayers is crucial for supporting the authors' final conclusions. In the authors' response to the Reviewer #2's comment (1), I believe that relying solely on previous work as evidence is not sufficiently convincing. The authors can provide HRTEM images of the structure in the manuscript (if AC-STEM characterization is available, the evidence would be even more compelling), and present intuitive EDS spectral data to corroborate the presence of the Pd monolayers.

> We now provide TEM images of Au@Pd NPs in Fig. S1 which have a Pd adlayer on the Au NP surface. TEM images and EDS maps show even distribution of Pd atoms around the NP surface, without significant changes in morphology and size of original 80nm Au NPs, indeed suggesting Pd at the surface and Au in the bulk. However for such large NPs, even HRTEM cannot confidently assign the top atomic monolayer.

Thus in addition to TEM analysis, we carry out electrochemical cyclic voltammetry (CV) measurements which chemically demonstrate the existence of a Pd monolayer on Au NPs. In new Fig. S2 (shown as Fig. R1 below), CVs of Au NPs with different Pd coverage show characteristic oxidation peaks for Pd and Au. We note that the oxidation peak of Au completely disappears upon formation of Pd 1ML,

confirming that there are no pinholes. At the same time the 1ML PdO_x peak clearly appears at 0.42V, and this Pd oxidation peak shifts towards negative potential with increasing number of layers. This is in line with the CV results of Au@Pd films shown already (Fig. S21b). This confirms in both cases that we can create single 1ML Pd by UPD on Au. The oxidation peaks of Pd on Au planar films are better distinguished than on Au NPs due to the homogeneity of surface sites. This confirmation of these Pd monolayers is now discussed in the manuscript on page 3.

Fig. R1 | Cyclic voltammograms of Au (0ML Pd) and Au@Pd coatings for (a) nanoparticles (Fig. S2) and (b) planar films (Fig. S21b). Pd coverages are 0, 0.5, 1, 2 and 10ML (bottom to top). Scan rate = 10 mV/s, 0.1 M H₂SO₄. NPs are centrifuged and redispersed in water twice to remove extra capping ligands. Precipitated NPs are dropcast onto a glassy carbon electrode (working electrode) and then dried for electrochemical measurements. A Pt wire and Ag/AgCl are used for counter and reference electrodes, respectively.